

# Climatic niche shift and possible future spread of the invasive South African Orchid *Disa bracteata* in Australia and adjacent areas

Kamil Konowalik[1] and Marta Kolanowska[2,3]

[1] Department of Plant Biology, Institute of Biology, Wroclaw University of Environmental and Life Sciences, Wrocław, Poland

[2] Department of Geobotany and Plant Ecology, Faculty of Biology and Environmental Protection, University of Lodz, Łódź, Poland

[3] Department of Biodiversity Research, Global Change Research Institute AS CR, Brno, Czech Republic

Corresponding author
Kamil Konowalik,
kamil.konowalik@upwr.edu.pl

## ABSTRACT

Orchids are generally regarded as plants with an insignificant invasive potential and so far only one species has proved to be harmful for native flora. However, previous studies on *Epipactis helleborine* and *Arundina graminifolia* indicate that the ecological aspects of range extension in their non-native geographical range are not the same for all species of orchids. *Disa bracteata* in its native range, South Africa, is categorized as of little concern in terms of conservation whereas in Australia it is naturalized and considered to be an environmental weed. The aim of this research was to determine the ecological preferences enabling the spread of *Disa bracteata* in Western and South Australia, Victoria and Tasmania and to evaluate the effect of future climate change on its potential range. The ecological niche modeling approach indicates that most of the accessible areas are already occupied by this species but future expansion will continue based on four climate change scenarios (rcp26, rcp45, rcp60, rcp85). Further expansion is predicted especially in eastern Australia and eastern Tasmania. Moreover, there are some unpopulated but suitable habitats in New Zealand, which according to climate change scenarios will become even more suitable in the future. The most striking result of this study is the significant difference between the environmental conditions recorded in the areas which *D. bracteata* naturally inhabits and invasive sites—that indicates a possible niche shift. In Australia the studied species continues to populate a new niche or exploit habitats that are only moderately represented in South Africa.

## INTRODUCTION

The study of biological invasions has been called "one of the hottest current topics in ecology" (*Sol, 2001*), mostly because together with habitat destruction and climate change the spread of non-native organisms is considered to be a major threat to biodiversity. In Australasia, invasive (= not native) species are a major problem; for example, the number

of species of plants reported as introduced, that have been released and may or may not have become naturalized in Australia exceeds 28,000 (*Randall, 2007*). A subset of them have naturalized and are a threat to the rich endemic flora of that continent (*Coutts-Smith & Downey, 2006*; *Randall, 2007*; *Duursma et al., 2013*) as the populations are self-sustaining and spreading without human assistance. In addition, the eradication of invasive weeds is costly (*Sinden et al., 2004*). Governmental agencies and private landowners invest large amounts of money in controlling the spread of weeds using various methods to assess risk of their further spread. One of the emerging techniques is to evaluate the potential future ranges of invasive species using climatic niche modeling (*Peterson, 2003*).

In this research we implemented modeling approach to evaluate the possible further spread of invasive orchid species, *Disa bracteata* Sw., which was first reported in Australia relatively recently, in 1944. This plant is listed in the Global Compendium of Weeds (GCW; http://www.hear.org/gcw) and is the only weedy representative of the mainly sub-Saharan genus *Disa* P.J. *Disa bracteata* is classified in the GCW as an environmental weed (species that invade native ecosystems; (*Blood, 2001*) or a naturalized species (self-sustaining and spreading populations but not necessarily affecting the environment; *Barker et al., 2005*; *Hussey et al., 1997*). *D. bracteata* is a South African endemic plant found in both the Eastern and Western Cape (*Foden & Potter, 2005*), where it is widespread and common, especially in areas subject to mild disturbance. In undisturbed vegetation it is somewhat less frequent. *D. bracteata* is included on the Red List of South African plants as a taxon of Least Concern (*Raimondo et al., 2009*; *Foden & Potter, 2005*). In the mid-20th-century it was brought to Australia where it became naturalized (*Groves et al., 2003*). This orchid was first formally recorded near Bacchus Marsh, west of Melbourne, in 1944 (*Wileman, 2015*; *Land Management Team, 2015*) and since 1945 additional reports came from areas of the Great Southern Region in Western Australia (vicinity of Albany). Later it was also recorded in South Australia (in 1988) and Victoria (in 1994). Recently it was found in Tasmania (Viridans Biological Databases). In Australia *D. bracteata* was probably accidentally introduced and it is now growing along roadsides. Invasive populations are large with up to almost 80 mature individuals in one square meter (*Tucker, 2006*; Trees For Life).

The aim of this research was to evaluate the similarities in the bioclimatic niches occupied by invasive and native populations of *D. bracteata* and to estimate the potential further spread of this species in Australia and adjacent areas. The comparison of the bioclimatic conditions experienced by African and non-native plants was conducted to explain the nature of this invasion by exploring the possible changes in the bioclimatic preferences of *D. bracteata* in early and present stage of its spread in Australia. Currently it is unclear whether the studied species inhabits similar climatic niches in Australia and Africa or whether it was able to colonise new niches in Australia. The negative effect of the introduction of exotic orchid on the native flora was reported only once so far (*Recart, Ackerman & Cuevas, 2013*), but the actual impact of invasive Orchidaceae on local plant communities remains poorly recognized. We do believe that results of our analyses of possible future spread of *D. bracteata* will be valuable information that can be used in planning conservation actions.

## MATERIAL AND METHODS

### Localities

A database of localities of *D. bracteata* was prepared based on the information recorded on the labels of identified herbarium specimens deposited in MO, WAG, S, NY, AD, MEL, CANB, HO, NSW, and PERTH. The herbaria acronyms follow *Index Herbariorum* (*Thiers, 2018*). The process of georeferencing follows *Hijmans et al. (1999)*. The geographic coordinates provided on the herbarium sheet labels were verified. If there was no geolocation data on the herbarium sheet label, the description of the collection place recorded was assigned coordinates as precisely as possible. In addition, the information provided by the South African National Biodiversity Institute and Global Biodiversity Information Facility (GBIF) with a precision value of less than 1,000 m was used. A total of 187 native (1863–2013) and 747 invasive records (1945–2016) were gathered (File S1).

### Niche modeling

The terminology used in this paper follows *Peterson & Soberón (2012)*. The distribution of the studied orchid was evaluated by including in the analyses numerous parameters potentially relevant to its occurrence. While the bioclimatic variables were commonly used as the only predictor in previous studies on invasive plants (e.g., *Mainali et al., 2015*; *Wang et al., 2017*), in our research we also incorporated vegetation, soil, and topographic factors. Due to the lack of data on distribution of pollinators of *D. bracteata* and insufficient information on the mycorrhizal associations of this species these two ecological aspects were omitted in the analyses.

Nineteen bioclimatic variables from the CHELSA version 1.1 database (*Karger et al., 2016a*; *Karger et al., 2016b*) were used. The recent study by *Bobrowski & Schickhoff (2017)* indicated that this dataset performs better than other available climatic data in ecological niche modeling. Eighteen soil characters relevant to plant growth were obtained from Global Soil Information Based on Automated Mapping (*Hengl et al., 2014*; http://www.soilgrids.org) with a 250 $m^2$ resolution and upscaled to match the resolution and extent of the bioclimatic variables. Furthermore, several other georeferenced factors were used in the analyses: potential vegetation (*Ramankutty & Foley, 1999*), soil quality (*Fischer et al., 2008*) along with six topographic variables based on an altitude raster (File S2). Because some previous studies (*Barve et al., 2011*) indicated that usage of a restricted area in ENM analysis is more reliable than calculating habitat suitability on the global scale the region of our analysis was clipped using a rectangular mask enclosing known populations and surrounding regions in order to estimate possible migration and/or spread. Since this species continues to spread mainly in Australia, the northern border of the Australian continent was set as the maximum extent of spread in this study. To account for co-linearity and select the most important bioclimatic variables, the number of original bioclimatic data was reduced using the R package MaxentVariableSelection (*Jueterbock et al., 2016*). The following criteria were applied: correlation threshold was set at 0.7, contribution threshold at 1 and beta-multiplayer was tested in the range of 1 to 15 using 0.5 steps and in the range of 1 to 1.5 using 0.1 steps. For each setting, the model was run 10 times and the results were averaged to decrease the possibility of a random selection

even though all variables were treated *a priori* as equal. This algorithm evaluates correlation and contribution mutually and is more objective than selection based on an expert opinion and correlation which may be not repeatable and biased by the specific preferences of a researcher.

The modeling was conducted using the maximum entropy method implemented in Maxent version 3.3.3 k (*Phillips, Dudík & Schapire, 2004*), which is commonly used in ecological studies and is known to be reliable (*Kolanowska & Konowalik, 2014* and references therein). The maximum number of iterations was set at $10^4$ and convergence threshold at $10^{-5}$. For each run, 20% of the data was set aside and used as test points (*Suárez-Seoane et al., 2008*; *Konowalik, Proćków & Proćków, 2017*; *Walas et al., 2018*). The "random seed" option, which provides a random test partition and background subset for each run was used. The run was performed as a bootstrap with $10^3$ replicates and the output was set to cumulative.

To evaluate the possible future expansion of *D. bracteata* within Australia, climate projections obtained from Coupled Model Intercomparison Project Phase 5 (CMIPP5) were used. Four "representative concentration pathways" (RCPs: rcp26, rcp45, rcp60, rcp85), which differ in predicted $CO_2$ concentration (*Collins et al., 2013*), were analyzed. We only considered the models covering all four representative concentration pathways for the year 2070 (average for 2061–2080). These models were obtained from WorldClim (http://www.worldclim.org) (File S2). To reduce the bias caused by the selection of only one specific model, they were averaged and the ensemble map for each variable, in particular RCP, was computed (*Konowalik, Proćków & Proćków, 2017*). This step simplifies the interpretation as it shows the general trend specific for a given RCP scenario while reducing extremes and uncertainties of particular models. Since many soil variables may be potentially affected by climate warming and due to the absence of such models for the region studied they were not used for future predictions. For the presentation of results, output grids with a cumulative scale were converted to binary grids using maximum training sensitivity plus a specificity threshold (*Liu et al., 2005*; *Liu, Newell & White, 2016*).

A range of methods described in previous studies (*Kolanowska & Konowalik, 2014*) were used to analyse output from niche modeling and quantify differences between indigenous and invasive populations. To measure the degree of similarity between occupied niches of both groups the niche equivalency test (*Warren, Glor & Turelli, 2008*) was calculated using 'ENMTools' R package (*Warren, 2016*). Niche overlaps (D and I) were calculated using methods of *Warren, Glor & Turelli (2008)* and *Broennimann et al. (2012)*. Schoener's D statistic uses direct measures of species density, which in this study were changed to measures of densities of occurrence modelled in environmental space. 'I' statistic was based on the modified Hellinger distance that compares two probability distributions. These two metrics range from 0 (no similarity) to 1 (high similarity). The bias metric (*Pressey et al., 2000*) was calculated as described previously (*Kolanowska & Konowalik, 2014*). It shows differences in ecological tolerance between invasive and native populations and the value of this metric can be either positive (when novel conditions with higher values are experienced by invasive populations) or negative (if conditions recorded in indigenous populations have higher median than those in newly occupied areas). To visualize interdependence

of different populations in the simplified (two dimensional) space Principal Component Analysis (PCA) was performed on all available variables (Appendix 3), as described in *Kolanowska & Konowalik (2014)* using R (*R Core Team, 2016*). All GIS operations were done in open source software QGis (*Quantum GIS Development Team, 2016*) and R (*R Core Team, 2016*) using packages 'raster' (*Hijmans, 2016*) and 'rgdal' (*Bivand, Keitt & Rowlingson, 2016*).

## RESULTS

### Variable selection and model evaluation

According to MaxentVariableSelection (*Jueterbock et al., 2016*), the lowest AIC, AICc, BIC scores and highest AUC for training dataset are assigned to the model with beta-multiplayer = 1. The final set of the most important and uncorrelated variables included eight of the original 49 variables: temperature Annual Range (Bio7), Mean Temperature in Wettest Quarter (Bio8), Precipitation Seasonality (Bio15), Precipitation in Warmest Quarter (Bio18), Precipitation in Coldest Quarter (Bio19), Sand Content, Soil Organic Carbon Content, Soil pH.

The calculated value of the area under the curve (AUC) was 0.98 (SD 0.001), which indicates excellent model performance. Additional summary statistics included the mean cross-entropy (mxe), which equaled 0.019 (SD 0.001) and root-mean-squared error (RMSE), which was 0.073 (SD 0.002). Both of these values were very low, which also indicate a high reliability of created models.

### Potential distribution of suitable niches under current climatic conditions

The model of the current distribution of suitable bioclimatic niches for *D. bracteata* was calculated based on all uncorrelated variables and a model in which only climatic factors were considered exclusively were visually congruent (Figs. 1A, 1B, 2A and 2B). The main difference concerns the transitional zone between the western and eastern Cape in Africa and the Nullarbor Plain in Australia. The first region was not indicated as suitable in the model based on climatic factors, whereas the second one was shown as suitable for this orchid in this analysis. Otherwise the difference may be seen in the scale of suitable habitat—while the general trend is very similar, both models differ slightly in the extent of the predicted suitable niche. Currently, the species is found in most of the suitable habitats in both Africa and Australia. Yet there are some areas in Australia that are not colonized—the Eyre Peninsula and some smaller areas detached from the main distribution. A vast area of Tasmania was predicted to be suitable but so far the occurrence of *D. bracteata* has been reported only from the northern part of the island. Also, there were no records of this orchid from New Zealand (especially the North and South Islands) although the models indicate the existence of suitable habitat in this area.

### Future changes in the extent of suitable habitat

Future climate scenarios indicate that the extent of the suitable bioclimatic niches in South Africa will be very similar to the one observed today. Nevertheless, depending on the

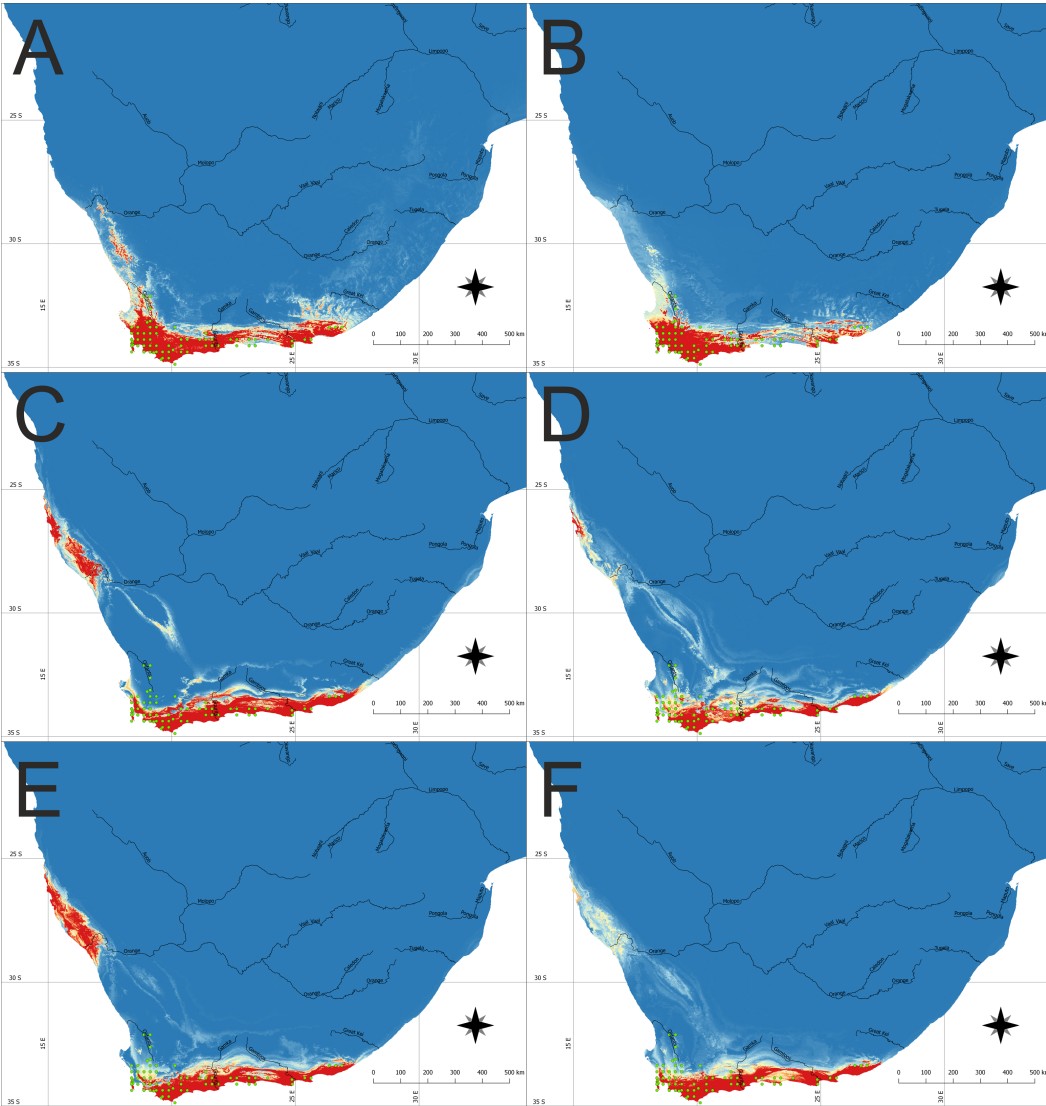

**Figure 1** **The potential area of the suitable niche for *Disa bracteata* in Southern Africa.** The insets visualize: (A) Potential niche modelled using current climate and soil variables, (B) Potential niche modelled using current climate variables, (C) Potential niche modelled using rcp26 climate change scenario, (D) Potential niche modelled using rcp45 climate change scenario, (E) Potential niche modelled using rcp60 climate change scenario, (F) Potential niche modelled using rcp85 climate change scenario. Blue indicates not suitable and red highly suitable. Green dots denote accessions used in ecological niche modeling. Lines show major rivers within the region. Maps were drawn using WGS 1984 (EPSG:4326) coordinate system.

climate change scenario used, the extent of native geographical range of the study orchid can slightly decrease or increase. Two models (rcp26 and rcp60) predict that suitable niches for *D. bracteata* will become available along the Atlantic coast, near the border of Namibia and South Africa, in areas around northern Namaqualand, Sperrgebiet and Lüderitz Bay (Figs. 1C and 1E).

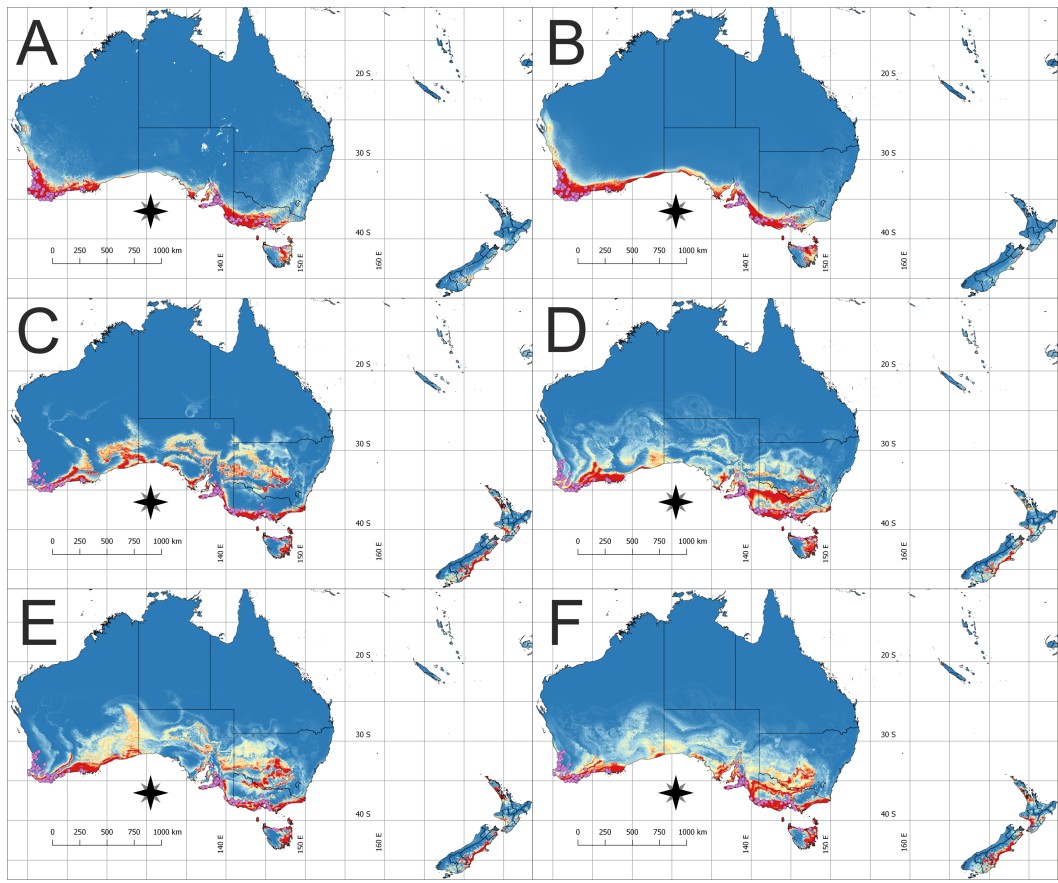

**Figure 2** **The potential area of the suitable niche for *Disa bracteata* in Australia and adjacent islands.** The insets visualize: (A) Potential niche modelled using current climate and soil variables, (B) Potential niche modelled using current climate variables, (C) Potential niche modelled using rcp26 climate change scenario, (D) Potential niche modelled using rcp45 climate change scenario, (E) Potential niche modelled using rcp60 climate change scenario, (F) Potential niche modelled using rcp85 climate change scenario. Blue indicates not suitable and red highly suitable. Purple dots denote accessions used in ecological niche modeling. Lines show administrative borders. Maps were drawn using WGS 1984 (EPSG:4326) coordinate system.

Within the invasive range, all models indicate that additional suitable niches could occur in New Zealand, especially along the southern coast of North Island and the South-eastern coast of South Island. Simultaneously, all scenarios predict range contraction in Western Australia, especially in the area north of Leeuwin–Naturaliste Ridge.

Apart from these general trends, each of the future climate scenarios gave specific information on the future potential range of *D. bracteata*. The rcp26 scenario prediction is that more suitable sites will be located in the south-western and south-eastern part of the Great Dividing Range, south-central and south-eastern part of the Nullarbor Plain, and in the northern part of New South Wales (Fig. 2C). Scenario rcp45 indicates a potential expansion of range in south-western and south-eastern part of the Great Dividing Range and south-western part of the Nullarbor Plain (Fig. 2D). Scenario rcp60, like the rcp26

scenario, predicts the occurrence of more suitable bioclimate in the south-western and south-eastern part of the Great Dividing Range, south-central and south-western part of Nullarbor Plain and in the northern part of New South Wales (Fig. 2E). Rcp85 scenario indicates that *D. bracteata* may spread in the south-western part of the Great Dividing Range and south-central and south-eastern part of Nullarbor Plain (Fig. 2F).

## Niche overlap and identity

The overlap between the studied environmental requirements of invasive and the natural populations is moderate or even low. Statistics calculated according to *Broennimann et al. (2012)* were $D = 0.35$ and $I = 0.58$. These values were slightly higher using the method developed by *Warren, Glor & Turelli (2008)*: $D = 0.44$, and $I = 0.73$. Also, niche identity tests reveal that the bioclimatic niches occupied by invasive and natural populations were different ($p < 0.01$). Yet visualization using PCA (Fig. 3) indicates a large overlap between the niches with only a small proportion of which is different.

Analyzed environmental conditions recorded at the sites occupied by this species are illustrated in Fig. 4. To measure the dissimilarity between invasive and native populations a bias metric was calculated. It indicates that all invasive populations occupy relatively different habitats compared to populations in Africa (Chi-squared $= 46.031$, $p = 4.9 \times 10^{-4}$). The same result is obtained when the invasive records are considered as one dataset or when the records are divided into those for Eastern and Western Australia (Chi-squared $= 73.807$, $p = 4.9 \times 10^{-4}$). Nevertheless, the suitable niches of both groups of Australian populations are also different (Table 1). There are, however, two similarities in the site characteristics of these two Australian regions: there is a greater sand content and lower precipitation in the warmest quarter in the areas occupied relative to that in areas occupied by natural populations. Others have either a medium difference (e.g., soil pH) or a significant dissimilarity (e.g., temperature annual range, mean temperature in wettest quarter, precipitation seasonality, precipitation in coldest quarter and soil organic carbon content; Table 1).

## DISCUSSION

Orchids are usually not regarded as weeds although some, e.g., *Epipactis atrorubens*, *E. helleborine* and *Dactylorhiza majalis*, colonize secondary habitats in temperate Europe (*Adamowski, 2006*; *Rewicz, Kołodziejek & Jakubska-Busse, 2015*). Even fewer species are reported as invasive (e.g., *Ackerman, 2007*) and so far only one, *Spathoglottis plicata*, has been shown to negatively affect native plants (*Recart, Ackerman & Cuevas, 2013*).

The invasive success of *D. bracteata* has not been thoroughly investigated and the mechanisms of this phenomenon remain unclear. The only predictive models are in Weed Futures (http://www.weedfutures.net; *Duursma et al., 2013*). Most ground orchids usually do not have great potential for spreading because of their very specific ecological requirements. Orchidaceae often have specific insect pollinators and specific mycorrhizal associations that need to be present in the soil to enable seed germination (e.g., *Batty et al., 2002*; *Cozzolino & Widmer, 2005*; *McCormick & Jacquemyn, 2014*). Many species of orchids will only germinate with the aid of one or a few species of fungus, so their distribution,

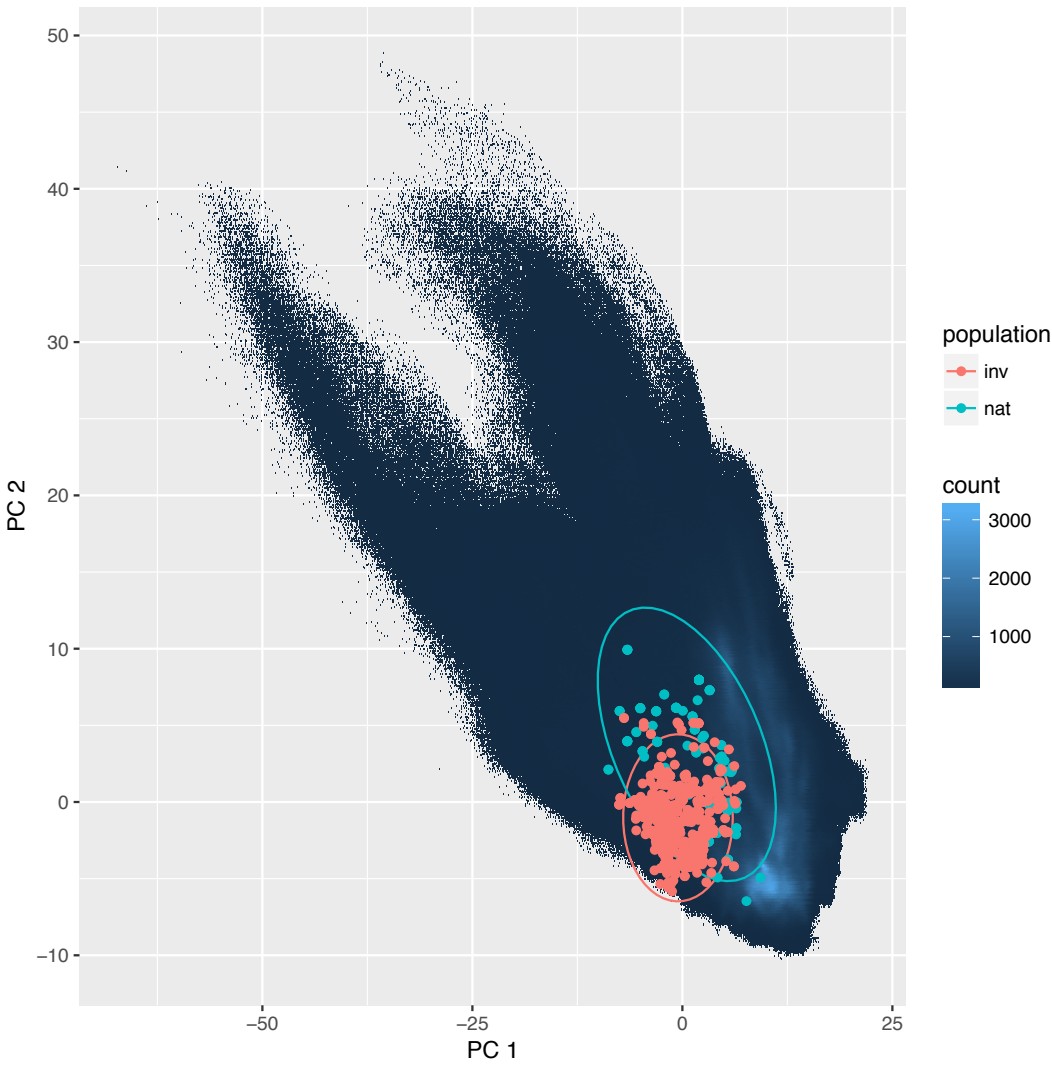

**Figure 3  Environmental niche of *Disa bracteata* as visualized by principal component analysis (PCA).**
Diagram was constructed with environmental values recorded for natural and invasive populations. The
blue background indicates the whole environment included in the analysis present in Africa and Australia.
Native and invasive populations are enclosed by circles encompassing 95% of the data. While native popu-
lations occupy a slightly broader niche some of the invasive populations occupy habitats not present in its
native niche.

and hence ecological success, is heavily dependent on suitable conditions for the fungus.
*Disa bracteata* appears to be able to form an association with a large number of fungal
partners, especially those that can survive in disturbed soils (*Bonnardeaux et al., 2007*),
thus it is much less limited in terms of the places and conditions in which it may become
established. For this reason, we did not incorporate the distribution of mycorrhizal fungi
in our analyses. It is noteworthy that the incorporation of a fungal factor in any analysis
of orchid distribution is extremely difficult. Most orchid mycorrhizal fungi belong to the
genus *Rhizoctonia*, a diverse polyphyletic group that is difficult to classify and molecular
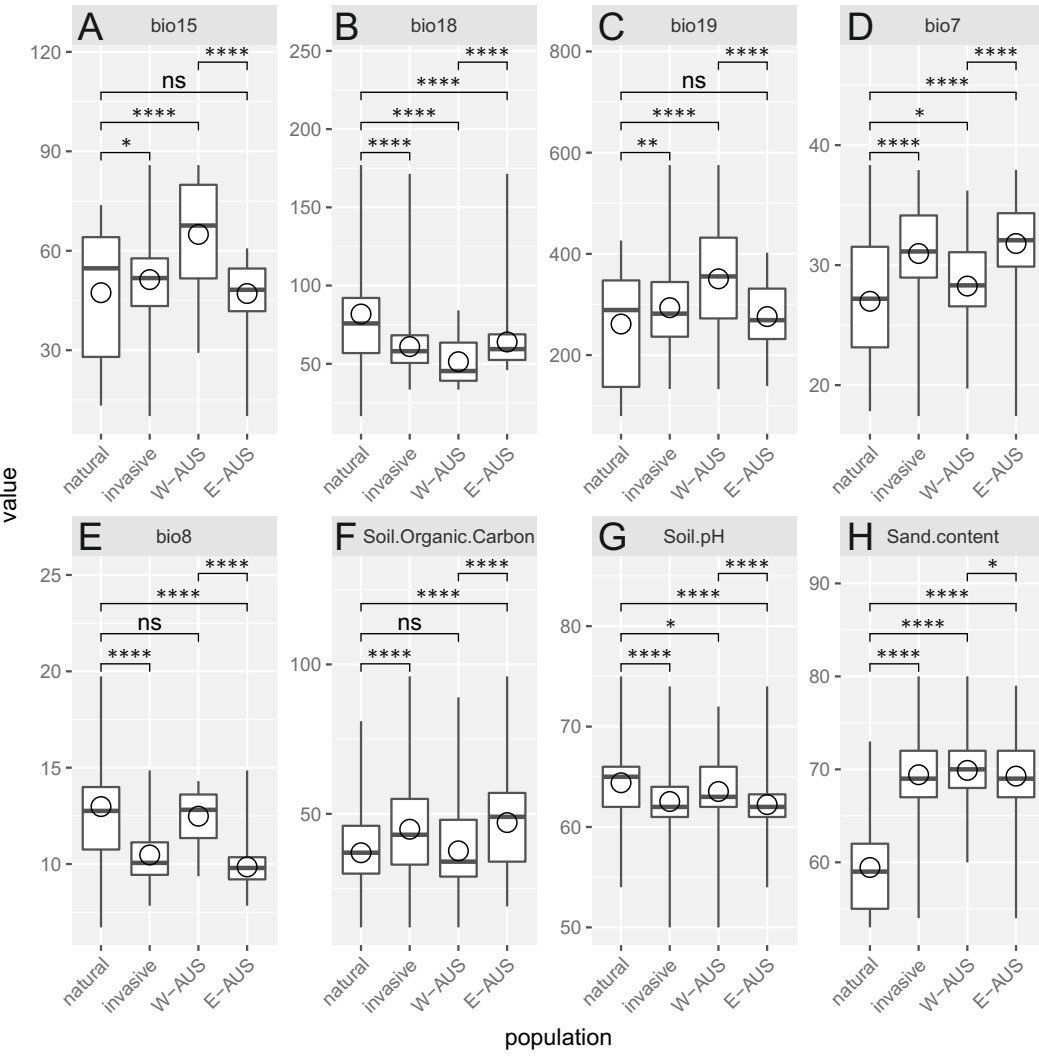

**Figure 4  Boxplot diagram of the environmental values recorded in the areas of occurrence of natural and invasive populations of *Disa bracteata*.** In addition to examining invasive populations as a whole they are divided into Western Australia (W-AUS) and Eastern Australia (E-AUS). Bio15 –Precipitation Seasonality (CoV), bio18 –Precipitation in the Warmest Quarter (mm), bio19 –Precipitation in the Coldest Quarter (mm), bio7 –Temperature Annual Range (°C), bio8 –Mean Temperature in the Wettest Quarter (° C), soil.organic.carbon content is expressed in (g per kg), Soil.pH refers to a pH ×10 in $H_2O$, Sand.content is expressed as a mass fraction in percent. Areas are compared to each other using *T*-tests. Circles are means, horizontal lines minimum and maximum values, the box represents first and third quantiles, and vertical line inside delineates the median.

methods have become the standard means of assigning these orchid fungi to groups within the *Rhizoctonia* alliance (*Bonnardeaux et al., 2007*). Because of the lack of data on the distribution of specific fungi it is not possible to use such data in ecological niche modeling.

As seed production in *Disa bracteata* is pollinator-independent there was no need to incorporate the potential distribution of any pollinator in order to get a more realistic

**Table 1  Results of bias metric.** If invasive populations would occupy the same habitat as natural the result will be 0. Negative values indicate occupation of sites below median found in the natural range, whereas positive values indicate occupation of sites above the median. The greater the number is (or lower in case of negative values) the greater is the difference.

| | bio07 | bio08 | bio15 | bio18 | bio19 | Sand content | Soil organic carbon content | Soil pH |
|---|---|---|---|---|---|---|---|---|
| Australia | 24.9 | −23.6 | −6.0 | −10.5 | −7.1 | 47.6 | 14.7 | −14.3 |
| Eastern Australia | 27.9 | −24.9 | −8.8 | −10.3 | −8.0 | 47.6 | 16.0 | −14.3 |
| Western Australia | 6.2 | −4.0 | 11.9 | −12.9 | 10.5 | 52.4 | −2.0 | −4.8 |

potential distribution of this orchid. *Disa bracteata* appears to be self-pollinating as a result of the breakup of pollinia in the anther (*Kurzweil & Johnson, 1993*). Generally, this is not beneficial for genetic variability, however it does enable it to produce a large number of seeds. High propagule pressure greatly enhances the chances of the establishment of invasive species (*Colautti, Grigorovich & MacIsaac, 2006*). In fact, genetic variability seems to be less important as many successful invaders reproduce vegetatively. This is frequently the only mode of reproduction in some invasive species. This is often influenced by environmental conditions that are not suitable for the full development of a plant e.g., maturation of seeds. Ideally, even in such cases an invasive species may couple vegetative propagation with occasional sexual reproduction in order to respond to a suite of selective pressures and propagate efficiently (*Atwater et al., 2017*).

The environmental similarity between Australia and South Africa enabled numerous African plants to naturalize in Australia. 15% of naturalized flora of South Australia consists of species native to South Africa (*Kloot, 1986*; *Scott & Panetta, 1993*). Slightly higher contribution of African plants was observed in Western Australia (17%; *Scott & Delfosse, 1992*; *Scott & Panetta, 1993*). It is worth to notice that the two species considered as major environmental weeds in Australia and New Zealand, *Chrysanthemoides monilifera* (Asteraceae) and *Asparagus asparagoides* (Asparagaceae), are of African origin (*Thorp & Lynch, 2000*).

Invasive species often experience release from biotic interactions and dispersal barriers in their non-native ranges (*Torchin et al., 2003*; *Colautti et al., 2004*; *Jiménez-Valverde & Peterson, 2011*). While the visualization using PCA indicate a large overlap between the niches of African and invasive populations of *D. bracteata* they are not the same and that small difference is important as it influences the results of the niche identity test.

Another interesting result is that the niche of invasive populations has changed over time as the colonization process has progressed. At the time of the first introduction, which was around 1944 (date of the first recorded specimen), only a few localities were known and the conditions there were similar to those in its natural range. However, over time more populations were established, which eventually gave rise to the colonization of the eastern part of Australia. The first georeferenced specimen collected in 1989 was found on the southern Adelaide Plains. Dispersal to the western part of the continent involved colonizing novel habitats or those that are not available to *Disa* in its native range (File S3). This shift is congruent with the Köppen–Geiger climate classification system

(*Peel, Finlayson & McMahon, 2007*), which indicates that the difference in niches may be influenced by the climate availability or different preferences within both ranges. In Africa, six climatic types were occupied while in Australia two of them were not populated and the majority of established populations occurred within the Csb climate (Coastal Mediterranean). The pattern of shifting niche of invasive species was detected in some previous studies in cases when available evidence suggests use of novel environments by alien species in the invaded range (*Medley, 2010*; *Petersen, 2012*; *Di Febbraro et al., 2013*) when those conditions could be unavailable or inaccessible in the native range (*Broennimann & Guisan, 2008*; *Godsoe, 2010*; *Guisan et al., 2014*; *Qiao, Escobar & Peterson, 2017*). Additionally, there was a difference between Eastern and Western Australia: in the former second most frequent climate type is the Cfb (Marine With Mild Winter), while in the latter this position belongs to the Csa climate (Interior Mediterranean; Fig. 5). Thus it is not an entirely new climate but rather a shift of the climatic preferences that may be attributed to the establishment of invasive populations in areas possessing different composition of available climates. However, it may be related to novel preferences as well since all climates that are present in Africa are present in Australia. Interestingly, areas of some types of climate are less frequent in Africa and apparently are sparsely populated by *D. bracteata* but mainly by other *Disa s.l.* species. This may indicate that in its native distribution there are factors that prevent *D. bracteata* occupying this niche, such as biotic interactions (possibly with other closely related species) while in Australia this constraint is absent and *D. bracteata* is able to colonize these new sites. Studies on niche shifting species indicate this occurs in various areas (*Broennimann et al., 2007*; *Elith, Kearney & Phillips, 2010*) but it is relevant to fewer than 15% of plant invaders (*Petitpierre et al., 2012*). There are significant differences between the climatic niches occupied by invasive and native populations of another invasive species of orchid, *E. helleborine* (*Kolanowska, 2013*), but in this case, no shift within the Köppen–Geiger climate was found. A study of the niche shifts of plants introduced into Australia found that none of 26 species included in this study changed their Köppen–Geiger climatic niche (*Gallagher et al., 2010*).

In the case of *D. bracteata* a significant difference in the proportion among climates in areas occupied in Africa and Australia was recorded (Fisher's Exact Test $p = 1.5 \times 10^{-5}$). The same result is obtained when Africa is compared with two regions of Australia considered separately (Africa vs. Eastern Australia: Fisher's Exact Test $p = 1.4 \times 10^{-4}$, Africa vs. Western Australia: Fisher's Exact Test $p = 1.3 \times 10^{-3}$). This suggests that irrespective of invaded region, *D. bracteata* occurs in Australia in habitats characterized by different climatic conditions (in terms of climatic zones) than these recorded in Africa. Even though both contain the same climatic types, they differ in their abundance and relative areas. This may be influenced by the total available land area which is much smaller in Southern Africa that is stretched along meridians while Australia is rather stretched along parallels.

*Disa bracteata* may be the first record of such a shift within Australia. When estimating niche conservatism and changes in future range, one needs to bear in mind that correct estimation of the latter is tricky because acquiring a new niche is an active process possibly linked to novel adaptations that may not be known at the time of a study (*Thuiller et al., 2008*; *Elith, Kearney & Phillips, 2010*). *Alexander & Edwards (2010)* suggested that the

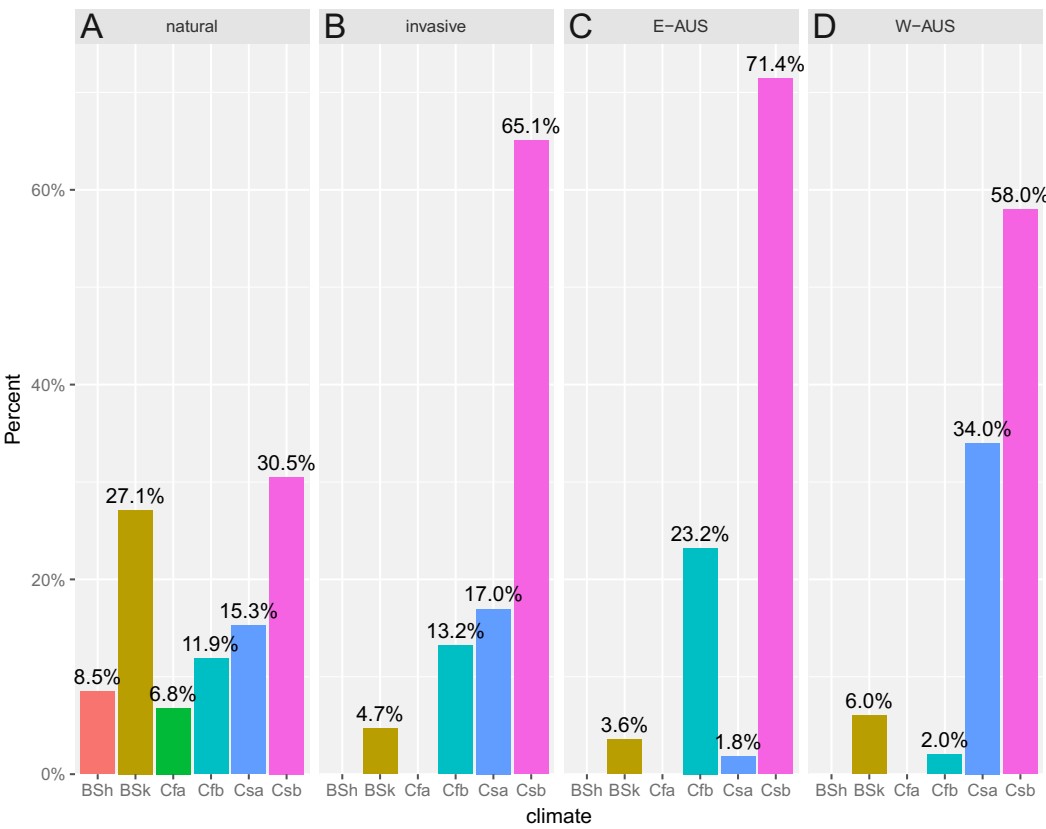

**Figure 5** **Histogram of climates recorded in the known populations divided into: (A) natural (South African), (B) invasive (Australian), (C) E-AUS (Eastern Australian), and (D) W-AUS (Western Australian).** Two types of climate present in Africa are not occupied in Australia. Invasive populations occur mainly within Csb climate with Cfb as the second most frequent in the Eastern Part and Csa as the second one in the Western Part of Australia. Shortcuts follow standard Köppen–Geiger climate classification system: BSh, Hot Semi-Arid; BSk, Cold Semi-Arid; Cfa, Humid Subtropical; Cfb, Marine with Mild Winter; Csa, Interior Mediterranean; Csb, Coastal Mediterranean. Prior to the analysis, occurrences were rarified to match resolution of climate map (10 km$^2$).

probability of a niche shift in invasive species depends primarily upon the ecological and genetic processes limiting the species in its native range. Unfortunately, during our studies we did not have access to a sufficient amount of molecular data to explore genetic differences between African and invasive populations of *D. bracteata*.

## CONCLUSIONS

A South African *D. bracteata* has become invasive in Australia and it is already present on a large part of the continent. Created models suggest that area with the suitable niche for this species is larger than that currently occupied by the studied orchid thus; the spread of this species will continue. How this expansion will proceed depends on future changes in the factors influencing its distribution and primarily on the magnitude of the climate modification. As demonstrated here, it is very likely that a niche shift has occurred in this case so the further spread of studied orchid should be monitored. Altogether, the results

of this study indicate the need of further research on the spread of *D. bracteata*, especially analyses of genetic differences between native and invasive populations.

## ACKNOWLEDGEMENTS

We would like to thank Anthony Dixon and Manasi Mukherjee for language correction of our manuscript. Nigel Swarts and other anonymous reviewers are thanked for their suggestions improving this manuscript.

### Funding

The research described here was supported by the Grantová Agentura České Republiky (GA ČR; grant nr 14-36098G). The funders had no role in study design, data collection and analysis, decision to publish, or preparation of the manuscript.

### Grant Disclosures

The following grant information was disclosed by the authors:
Grantová Agentura České Republiky: 14-36098G.

### Competing Interests

The authors declare there are no competing interests.

### Author Contributions

- Kamil Konowalik conceived and designed the experiments, performed the experiments, analyzed the data, contributed reagents/materials/analysis tools, prepared figures and/or tables, authored or reviewed drafts of the paper, approved the final draft.
- Marta Kolanowska conceived and designed the experiments, analyzed the data, authored or reviewed drafts of the paper, approved the final draft.

### Data Availability

The raw data are provided in the Supplemental Files.

### Supplemental Information

Supplemental information for this article can be found online at http://dx.doi.org/10.7717/peerj.6107#supplemental-information.

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
