# Peer review of "Climatic niche shift and possible future spread of the invasive South African Orchid Disa bracteata in Australia and adjacent areas"

_PeerJ, doi:10.7717/peerj.6107_

## Round 0.1 · original submission · Major Revisions

Thanks you for submitting this interesting paper. As you can see we have a range of views from the reviewers, most are around clarification although the 2nd reviewer raises important questions regarding the models you use and why. These would need to be address before the paper could be considered for publication.

Reviewer 1 ·

Basic reporting

There are several sections of the manuscript that could use judicious editing for clarity, including the Introduction and the Conclusions, which seem garbled in the first few lines.
And the authors seem to have an aversion the word "the." Many times while reading, I was halted in my tracks for it's lack. The judicious use of the article would greatly enhance clarity.

Specifically:

In both the abstract and the introduction you state that Disa bracteata is considered an environmental weed or a naturalized species. Weeds are generally considered to be noxious while naturalized species or not. This is an important distinction. Please clarify what you mean. Is it a matter of where it occurs? Perhaps it is weedy in some places but not in others…

In the first paragraph “invasive,” “naturalized” and “weedy” are not well delineated. This could be clarified in the manner provided below:

Line 38 “In Australasia introduced species are in essential problem and the number of introduced plants exceeds 28,000. The subset of them have become naturalized… A smaller subset have become invasive, and the expansion of exotic weeds incurs significant costs related to their eradication.“

Line 43 to 44 should read “control the spread of certain species and use various methods to undertake risk assessment“

Line 45 “trough” should be “through”

Line 47 should read “...to study the case of the invasive orchid species ...“

Line 50 through 53 the species is classified as an environmental weed or naturalized species based on what? Does it depend where the species is found? This needs to be better defined.

Line 66 through 67 please specify where you are conducting your research the title states Australia, which would be in the invasive range of the species.

Line 104 to 105 clarify that 20% of the data or set aside to be used as test points.

Line 122 please clarify the variables used for your principal components analysis

Line 130 to 132 should read “the final set of the most important and uncorrelated variables included eight of the original 49:”

Line 135 Should read “The calculated value of the area under the curve”

Line 142 should read “...uncorrelated variables and model in which climatic factors were exclusively analyzed...”

Line 143 should read “The only difference concerns the transitional zone ...“

Line 144 “between the western and eastern cape in Africa, and the Nullarbor Plain in Australia”

Line 145 should read “was not indicated as suitable in the model based on climatic factors, whereas the second one was shown as suitable…”

Line 161 change and invasive range to the invasive range

Lines 183 to 192 beginning with another interesting result… To the end of the paragraph. This section seems more suitable to the discussion then to the results section

Line 210 there are invasive or kids in both Fiji and Florida. The ones in Fiji are also terrestrial, though they occur in forested habitats.

Line 211 should read “The invasive success of D. bracteata has not been thoroughly investigated, and the mechanisms of this phenomenon remained unclear.“

Line 216 change fungi to fungal

Line 224 should read “Generally, this is not beneficial for genetic variability, however it is advantageous to produce a large number of seeds.”

Line 227 should read “seems to be a less important trait as many successful invaders are characterized by an efficient vegetative reproduction. This is frequently the only mode of reproduction for some invasive species. Ideally, even an invasive species will couple vegetative propagation with the occasional sexual reproduction to add the ability to respond to a suite of selective pressures.”

Line 234 delete "for example"

Line 244 capitalize south

Line 261 “In the case of Disa there seems to be a significant difference between the climate of occupied habitats in Africa and Australia. When Africa and Australia are compared, the latter split into two groups.”


Please clarify lines to 267 through 269. I think what you were trying to say is that when a new niche is occupied this is an active process and maybe linked to novel adaptations which may not be discovered during the study itself.

The conclusions need to be entirely reworked, particularly the first two lines. I'm not entirely sure I understand what you are trying to convey here.

Experimental design

I thought the methods used were appropriate, and well substantiated by the citations included.

I would like to have seen what variables were used in the Principal Components Analysis.

Also, the range maps are difficult to read, with the model and test points indistinguishable from the background. This makes it impossible to visually evaluate model fit.

Cartographically, they are inadequate. There is no compass rose or other directional indication. In the South African models, the geography is indistinct, with no attempt to clearly identify the geographic boundaries. There is no context for the reader to evaluate the models.

Validity of the findings

I think that the conclusions are valid - however, the way they are presented in the text is a bit confusing. Some of the text in the Methods section seem more appropriate to the Discussion, as noted above.

The range/habitat results should be clarified - as presented in the text they are a bit muddled - especially because no attempt is made to identify the regions they indicate on the models they present. In other words, there is no cartographic evidence available to evaluate their interpretation of the models.

Also, the inference of a niche shift in the invasive range seems intriguing, but the supporting evidence is not well established. In particular, a better explanation of the results of the Exact Test, and the bearing this has on the Koppen-Geiger shift, should be undertaken.

Reviewer 2 ·

Basic reporting

The English needs to be improved.
The aims and hypotheses need to be stated - the basis of the study was not clear.
I provide detailed comments below.

Experimental design

The research question was not well defined.
The methods were appropriate but their use was ambiguous.
I provide detailed comments below.

Validity of the findings

The data were robust, although their interpretation was unclear.
The conclusions were not well stated.
I provide detailed comments below.

Additional comments

This manuscript investigated the environmental niche of Disa bracteata, one of the few orchid species to become potentially invasive in Australia outside of its native range, South Africa. For this, the authors compiled a database of all known records for D. bracteata and used a Maxent framework to model its niche under present day conditions and under different climate change scenarios. The authors used a variety of environmental variables and constructed reliable niche models of the distribution of D. bracteata. Overall, I thought that the study was interesting in that D. bracteata is a “weedy” orchid and that climate change might influence where it could occur in future. However, this manuscript has a number of limitations, most importantly that it lacks a proper research question. Also the link with climate change is unclear i.e. why study the niche of this orchid under climate change scenarios? As a result the implications of the study are unclear. There are numerous lapses in the writing and presentation of this manuscript, which would need to be improved. The ENM approach needs a better justification – why were so many variables included prior to the Maxent variable selection process. Not all variables are equal a priori. This would affect what variables are selected to begin with. I recommend a hierarchical method of variable selection whereby highly correlated variables are removed (>0.7), then examine the relationships between the remaining variables and see if they are ecologically relevant. Many soil variables were included, but no geological substrate variables. Including many soil variables would not necessarily improve Maxent performance. Geological substrate is an excellent predictor of plant distributions and should be included in niche models of all plants. The future climate change scenarios are not well explained, and the methods the authors used to estimate the niche of D. bracteata under climate change is unclear. The authors state that they averaged the climate change models for each RCP. How many models were included for each RCP? These future scenarios often differ in their underlying code and ability to predict environmental change, even within a given RCP. This does not mean there will be bias from one particular model as the authors claim, it would depend on the justification of using a particular model (or models). Therefore, averaging them is not appropriate if you want to make predictions on future influence of climate change on species distributions. They would have to be examined individually. I provide some specific comments on the text below.

Abstract
The abstract needs to be re-written. Environmental niche modelling has been used widely over the past 20 years to investigate the suitable habitat for many invasive and potentially invasive taxa. Why study Disa braceata? What do you mean by “in an aspect of conservation”? How was future expansion quantified? How many climate change models were used? The concluding statement is overstated – D. bracteata occurs over a wide range of conditions in South Africa and can self-pollinate and reproduce clonally. That it encounters similar habitat in Australia and may expand in future is not so profound.

L35: A citation from 2001 is not recent.
L38: This needs to be re-written.
L44-46: They are not a “promising technique” - the use of ENMs has been extensive in invasion biology for the past 20 years.
L47: The link with D. bracteata is not clear. Just because it is a “weedy” orchid, does not mean that it merits thorough investigation.
L86-87: What other data are you referring to here?
L66-68: What is the research question?
L77-78: What do you mean by “precisely placed on the map”?
L93-94: What do you mean by “clipping using a rectangular mask”?
L96: Add ‘for’.
L98-100: It is not clear what these steps mean or do to the data.
L103: There are more general references for the reliability of Maxent in the literature.
L108: The jump to climate change scenarios is not clear.
L111-115: State exactly what models under CMIP5 you used. Did you use every model? While some future scenarios are similar (e.g. CSIRO and HAD), each model is different in its assumptions and underlying code. Therefore, aggregating them and averaging their influence on a species’ niche under each RCP may obscure this.
L115-116: What are “continuous grids”?
L117-125: More explanation is needed here. This is presumably to test for the differences between South Africa and Australia. Do you also use niche equivalency to test for differences between current scenarios and future climate change? What is a “bias metric”? Why was a PCA used here?
L176-205: You should explain better in the methods section what each of these statistics calculate. Then in this results section outline only the results. Some of the speculation here belongs in the discussion section.
L208: Orchids are often “weedy” in the sense that they often occur in ephemeral habitats. What you mean is that they are not normally classified as invasive.
L213-217: Many orchids automatically self-pollinate and mycorrhizas are generally ubiquitous in the soil. It depends on the species whether they have adapted to particular mycorrhizas.
L224:-232: This section needs to be re-written. There are a lot of different concepts introduced that are not well-epxlained.
L265: Can human introduction into Victoria be discounted?
L271-277: This conclusion needs to be re-written.

Reviewer 3 ·

Basic reporting

April 17th, 2018
Dear Prof. David Roberts
PeerJ Editor
The manuscript “Climatic niche shift and possible future spread of invasive South African orchid Disa bracteata in Australia” explores the potential expansion and habitat shift of an herbaceous weed. The authors have selected for this study an unusual orchid species that, unlike most species in the genus and the family, is of concern as an invasive alien plant in Australia and neighbouring islands. I trust this manuscript, if accepted for its publication in PeerJ, can contribute to a better understanding of how biological invasions evolve, providing to the readers of this journal a novel and more dynamic insight on plant invasions, by considering not only the current climatic scenario, but also edaphic parameters and some putative climatic scenarios to occur.

Despite of these positive aspects, I also have found some issues that should be clarified or improved.

Firstly, I suggest a modification of the title. This should reflect in a better extent which are the results here obtained. Contrary to the expectations, even when only Australia is mentioned in the current title, the manuscript explores the putative change in distribution of this plant in its native range, and in a broader invaded range (which includes Australia but also other neighbouring big islands). Indeed, this change could increase the interest of a wider public to read the paper because it covers a broader geographic scale.

Additionally, the English grammar along the manuscript is sometimes inaccurate, particularly the repeated omission of some articles, conjunctions and prepositions, which can hinder the understanding of the text. Other minor issues are reported in the list below.
In summary, I recommend this manuscript to be published in PeerJ, but only after the issues here mentioned are addressed by the authors, especially the two above mentioned. Besides, I am deeply interested in receiving the appropriate feedback on the authors’ criteria, and the actions derived from this and other reviews of the manuscript.

Experimental design

In this work, the authors assessed the potential ability of this particular plant to modify its habitat requirements in the invaded area, and they used this information to validate its potential distribution under different hypothetical climates to come. To this end, the authors performed a set of analyses that to my judgement are appropriated and sufficient to achieve their goals. In addition, they selected for this aim a very reliable source of information, including Chelsea, which resolution for this kind of analyses is more relevant that other alternatives (e.g. BioClim). In my criterion, there is not a clear justification for excluding the edaphic variables in the estimation of the potential distribution of this species under the assumed future climatic scenarios. I do not find a convincing explanation in the text about this. Curiously, the authors firstly included a group of edaphic variables, and in fact, later they found that three of these non-climatic variables are even crucial for the best-supported model. These variables were part of the set of most important and less-correlated variables to explain the distribution of this orchid weed. Nevertheless, the edaphic variables are simply omitted from the analyses that were based on the hypothetical future climatic scenarios. In this sense, I believe that there is not a clear explanation about why this procedure was followed. According to the authors when the figures 1 and 2 are explained, the suggested distributions following the model that includes and the one that excludes the edaphic variables are very similar (see the figures above mentioned). I assume this could be the explanation, but this is not obvious to me, and on the other hand this a not a solid justification for such procedure. In my opinion, that explanation is a very subjective statement, because in my criterion there is a certain area in each case for which the chance of occurrence of this species differs between the compared models (which occur for both the native and the invaded zones).

Validity of the findings

I consider the impact and novelty of the manuscript could be high due to the need to control this weed which expansion could have consequences not well understood for the local orchid floras in the invaded area. Data is robust, and its manage and analysis appropiate. The issue about the ignoring of the edaphic variables should solved, by including them or explaining in detail the reasons why they were not considered in the rest of the analyses (see comments). The discussion is in general appropiate but it could be improved by being a little less categorical in some points.

Additional comments

REVIEW OF THE MANUSCRIPT “Climatic niche shift and possible future spread of invasive South African orchid Disa bracteata in Australia”

General comments:

Please consider to include a broader geographic reference in your title. Note that your analyses are focused not only in Australia, but also in a broader range of the invaded region, and also in the native range of this plant. For me, maybe the aspect of higher concern is the next point. In my criterion, if the edaphic variables were found as being important in explaining the current distribution of this plant (which is explicitly mentioned in the manuscript, when you report the best fitted model, and all the variables that were analysed), then these edaphic variables should also be considered in the estimation of the potential distribution of this species under future climatic scenarios. I think it is biased to estimate changes in the species range for the next decades when the group of the edaphic variables is ignored. I assume that these variables will not change substantially in time, in the same time scale clearly. If this is not sure, please provide references supporting the exclusion of the edaphic parameters from the analyses.
In the Discussion you should explain that, considering this is one of the few orchid species completely pollinator-independent for seed production, there is no need to incorporate the potential distribution of any pollinator to get a more or less realistic potential distribution of the orchid. In contrast, the role of mycorrhizal fungi in this case is probably so determinant for this species as for most orchids. Nevertheless, you also mention that in the particular case of this species, the fungi are probably less limiting due to the use of a broad spectrum of fungi as mycorrhizal, or due to the exploitation of one particular or a few fungi species that are very abundant and well-distributed in its range. A short comment about how feasible or difficult is to obtain data about this kind of fungi to estimate the orchid distributions could also be mentioned (with the needed references). These two topics could enhance the relevance of your work by explaining why there is no need to consider the mutualistic partners of the orchid in the analyses you performed, and why the inclusion of data from the fungi should be considered in further studies.
I think the language, although good in general, should be improved to gain in clarity and propitiate an easier comprehension of the text by a broad spectrum of readers. Specific issues in this sense are indicated below (texts to be removed or included are always between quotation marks).

Specific comments:

Line 31: I think you could clarify in this point if those South African marginal habitats are also occupied by the species of interest.

Line 38-39: I suggest to replace “In Australia invasive species are essential problem and number of introduced plants in Australia exceeds 28.000” by ““In Australia, invasive species are an essential problem, and the number of plants reported as introduced in this continent exceeds 28.000 species”

Lines 40: Please replace “threat to rich, endemic flora” by “threat to the rich and highly endemic Australian flora”

Lines 47-50: I am afraid some information is missing here. Perhaps D. bracteata is the only species occurring out of the Sub-Saharan range of its genus. If that is the case, or even if not, please consider to rephrase this sentence. The different parts of it are not well connected.

Suggestion: “In the present research, we applied this approach to predict the future distribution of the invasive orchid Disa bracteata Sw., which is listed in the Global Compendium of Weeds (site address) despite of being part of a genus confined to only the Sub-Saharan portion of Africa (please include a reference supporting this)”.

Please be sure of rephrasing “the case of invasive species”. This has no sense to me if you do not include the particles “an” or “the”.

Lines 55-60: These two sentences are not well connected. Maybe you should introduce Australia as the region where this species was reported by the first time as a naturalized orchid weed. After that, mention the situation of the non-native flora in Australia (including the invasive species).
Line 76-78: The information provided in these two sentences is essentially the same. Please simplify the text by including the GBIF localities as a part of all the locations that we used. You could use this or a similar phrase: “including GBIF localities with a precision lower than 1000 m”.
Line 98: Please replace “applying following criteria:” by “applying the following criteria:”
Line 105: Please include a comma after the world option.
Lines 111-113: Redaction here could be improved.
Suggestion: “We only consider the models covering all four representative concentration pathways for the year 2070 (average for 2061-2080). These models were obtained from www.worldclim.org (Appendix 2)”.
Line 114: What in each case? Could you be more explicit? I think this information is quite relevant to understand the methodogy.
Line 114: Something is missing between “in” and “particular”. The sentence have no sense to me in its current version.
Line 116 (end): Please cite at least a reference at the end of this sentence. You should cite a study where this method was implemented by the first time or at least recently implemented.
Line 119: Readers should be interested in understanding the means of these two letters. Please include a brief explanation about it.
Line 124 Please note that the use of the conjunction “or” introduces uncertainty in this context. What did you use “in each case”?
Line 131: Please replace “eight from the originally included 49:” by “eight from the originally included 49 variables:”
Line 140: The distribution you are estimating is not the current one. It is the distribution estimated for the current conditions. Please, modify.
Line 140: Be sure the adjective “present” is modifying the noun “distribution”. Or it modifies “niche”?
Line 142: This has not sense to me. As I mention above, why do you use a model that ignores the edaphic variables, being those also important variables, as you demonstrate? This also applies for estimating the distribution of the orchid species under future climatic scenarios. In my criterion this a crucial issue. In this sense, please include in "Material and Methods" an appropriate explanation about why the edaphic variable were neglected.
Line 142: Please check if “considered” is more appropriate than “analysed”.
Lines 143-144: This congruence is not so clear to me. The occupation of a considerable area boarding the SE coast is suggested as very probable in A but not in B (red patches in between 7 and 8 o’clock in the panel A, but absent in panel B).
Line 150: Please provide any reference sustaining this statement.
Line 160 (end of): Please provide the exact panels when this information can be checked.
Line 161. Why “an” invasive range? You are not talking about a random invasive range. The expression "the invasive range" is, I think, more correct. Or better yet: "the potential invasive range".
Line 161: “Will” could be interpret as too categorical. I think “could” is a better option here.
Line 163: “the loss” instead of “loss”
Line 164: “the Northern area of” or “the North of...”
Line 165: Please guide the reader to find the panels of interest in each moment. Keeping the correspondence between all the scenarios and the panels could be a little difficult and frustrating without the inclusion of this information.
Line 175 (end): Please read the previous comment.
Line 177: “the ecological requirements”... instead of “ecological requirements”
Line 177: Rephrase this sentence by including the article “the” where it is required in the expression “the invasive and the native populations...”
Line 177: “moderate” instead of “medium”
Line 181: “indicates” instead of “indicated”. I suggest to put the verb in present because this a result.
Line 183: The correct verbal form is “influences” because “it” is third person.
Line 183: Please consider “of niche identity.” or “of the niche identity test.” instead of “of niche identity test”.
Line 184: Replace “is” by “could be”. The results in this kind of studies are estimations and therefore are not necessarily categorical.
Line 196: “habitats” instead of “habitat”
Line 200: “also different” instead of “different”
Line 201: “these two…” instead of “the two…”
Line 209 Please check for an extra space between the words helleborine and Dactylorhiza
Line 209-210: Please consider to include some examples from the tropics, where the role of alien orchids is not so well known but the area affected is proportionally much larger. Certainly, some of these invasive species are not exactly weed. For example, Eulophia alta, Oeceoclades maculata and Spathoglottis plicata can reach high densities even in not so disturbed areas.
Line 212: “phenomenon has remained” instead “phenomenon remained”.
Line 214-217: Please provide some references regarding these ideas.
Line 227-229: Consider to explore if there is any previous study documenting that the existence of a pollinator independent breeding system can provide a higher number of generations just occurring in a particular habitat (i.e. without interchanging genes from other populations under different climatic conditions). Theoretically, this could allow a faster local adaptation to the particular conditions of each site.
Line 244: “South” instead of “south”
Line 261: Does this apply to the genus Disa or to the species of interest? (which I consider is the point in this phrase). Please clarify.
Fig. 1 (Figure capture): The expression “for the orchid Disa bracteata” or something similar should be included. Each figure needs to be self-explanatory.
Fig. 1 (Figure capture): You need to distinguish more clearly between “current climate” and “future climate”. So, mentioning “climate” at the beginning for the panels A and B is not very accurate.
Fig. 1 (Figure capture): This is a question I made in many places along the manuscript. Why you did not consider the soil parameters in addition to the different future climate scenarios? Please provide this explanation in the section of Material and Methods of your manuscript.
Note: All comments above mentioned also apply for Fig. 2 as well. In addition, please note that dots in Fig. 2 representing the localities that were sampled should be green for simplicity, but they are in violet. Consequently, please modify the text or the images to keep the needed correspondence.
Fig. 3 (figure caption): Firstly, there is no reason to keep both sentences as separated paragraphs. Additionally, the text introducing this figure should not just describe what the reader can see by her/himself. The PCA shows much more information than the one provided in the text (e.g. 1. Degree of segregation in the abiotical niche of the native and invasive populations of Disa bracteata). Also, please provide more information about how you obtained the PCA.
Suggestion #2: “Enviromental niche of **** represented as a PCA resuming all environmental values recorded in natural and invasive populations of this orchid.”
Fig.4 (Figure Caption): Please check the comments for Figure Caption 3 and apply them to this one. What do these boxplots represent? I mean, what should be in the heading of your text? Both sentences should be joined in a single paragraph.
In addition: please also explain, as part of the caption text, the meaning of each climate variable code.

---

## Round 0.2 · Major Revisions

Thanks you for making the substantial corrections. Finding a reviewer/re-reviewer has been difficult so I apologize for the amount of time it has taken. We both like the manuscript and it is an interesting subject, however, more work is required as mentioned by the reviewer. Having read the manuscript and the rebuttal there are a number of places where you argue against the reviewer's review of the manuscript. While this is fine, I myself argue against reviewers (and editors, at the end of the day they are only our peers), you still need to incorporate the argument in some form into the text. Clearly, they have a difference of opinion or misunderstood something, this difference of option or misunderstanding will continue even if the manuscript where to be published. As such the manuscript still requires strengthening and the comments from the current and previous reviewers addressing in the text even if it is to prevent misunderstandings or support your opinion.

·

Basic reporting

The writing of this manuscript needs significant review. I have annotaed the document throughout with suggestions however at present there are some sentences and paragraphs which make understanding the main points difficult. Interestingly the written English in the rebuttel is significantly stronger than in the manuscript itself.
Whilst some of the important invasive literature is cited, I would have thought additional interpretive context with other invasive species studies could have been provided in the discussion

Experimental design

I do not profess to be an expert in modelling or in the methods employed for this study. Having read the rebuttel thoroughly and the earlier reviews by those who do seem to have more experience I do not think that the authors have responded to the comments adequately. Possibly the reason for this is that the data required to adequately understand drivers of distribution are simply not available in required detail.
I do not feel comfortable with the words 'ecological' or environmental' conditions being used to describe the bioclimatic variables used in this study.

I would prefer that the authors stick to 'bioclimatic' variables throughout as that is what is studies. I am also conscious of the word 'niche' which has quite braod connotiations. It should be prefaced with the word 'bioclimatic' because that was what was studied.

Validity of the findings

I think the findings are interesting and worth reporting however teh text needs considerable improvements beforehand

Additional comments

I am comfortable with the explanation provided by the authors not to consider the usual ecological drivers of orchid distribution - mycorrhizal and pollinator associations. I know the species well and these are certainly not variables easily modelled or likely to be of consequence. That said, I would have liked to see comparisons with similar studies on other non orchid invasive species.

---

## Round 0.3 · accepted · Accept

Many thanks for the corrections to your manuscript. I attempted to have the previous reviewers comment on the revision, but have been unsuccessful, hence the delay. However having read the manuscript I'm happy to recommend acceptance although I do note a number go spelling and grammatical errors that I hope you and the PeerJ staff will be able to pick up in the proof stage.

#